# Using Graphs to Perform Effective Sensor-Based Human Activity Recognition in Smart Homes

**DOI:** 10.3390/s24123944

**Published:** 2024-06-18

**Authors:** Srivatsa P, Thomas Plötz

**Affiliations:** Georgia Institute of Technology, Atlanta, GA 30332, USA; srivatsa@gatech.edu

**Keywords:** human-centered computing, ubiquitous and mobile computing, machine learning, smart-home, human activity recognition, pattern recognition

## Abstract

There has been a resurgence of applications focused on human activity recognition (HAR) in smart homes, especially in the field of ambient intelligence and assisted-living technologies. However, such applications present numerous significant challenges to any automated analysis system operating in the real world, such as variability, sparsity, and noise in sensor measurements. Although state-of-the-art HAR systems have made considerable strides in addressing some of these challenges, they suffer from a practical limitation: they require successful pre-segmentation of continuous sensor data streams prior to automated recognition, i.e., they assume that an oracle is present during deployment, and that it is capable of identifying time windows of interest across discrete sensor events. To overcome this limitation, we propose a novel graph-guided neural network approach that performs activity recognition by learning explicit co-firing relationships between sensors. We accomplish this by learning a more expressive graph structure representing the sensor network in a smart home in a data-driven manner. Our approach maps discrete input sensor measurements to a feature space through the application of attention mechanisms and hierarchical pooling of node embeddings. We demonstrate the effectiveness of our proposed approach by conducting several experiments on CASAS datasets, showing that the resulting graph-guided neural network outperforms the state-of-the-art method for HAR in smart homes across multiple datasets and by large margins. These results are promising because they push HAR for smart homes closer to real-world applications.

## 1. Introduction

Human activity recognition (HAR) in the context of smart homes has recently been regaining interest [1]. Two trends have been major drivers for the resurgence of interest: The first is the proliferation of inexpensive yet accessible, engaging, and helpful [2,3,4] smart home products such as Google Home and Amazon Alexa in a large number of households. Secondly, with a rapidly growing aging population [5], many applications have focused on the field of ambient intelligence and assisted living (AAL) technologies, which are aimed at improving the quality of life of seniors through the use of ubiquitous sensors [6]. HAR is one pertinent process in incorporating ambient intelligence in a smart home environment. It comprises modeling, reasoning, and decision-making procedures [7,8] with the aim of detecting and identifying complex human activities. Hence, improving the quality of life with AAL technologies depends on how well an interconnected network of sensors, which are capable of communicating and learning from user habits, can be used to detect and then identify a plethora of complex human activities in real-world settings. This can be achieved through processing collected spatial and temporal information [8,9].

A successful HAR system is one that is able to learn a user’s behavior during everyday life, often through a network of domotic sensors (e.g., motion sensors, etc.). The real-world settings of such a system might pose several challenges: (1) Variability—Sensors can stop working at any time, different sensors might record different values at different points in time, and observations of sensors might not necessarily be aligned in time [10,11]. This can be attributed to practical issues such as cost-saving measures, intermittent and unexpected failure of sensors, external forces in systems, etc. [12]. (2) Sparsity—Some activities occur rarely and/or activity occurrences may translate into a signal with a single value from one sensor. (3) Noise and redundancy—The same set of sensors may be triggered for different activities. (4) Limited data—It is challenging to collect large amounts of labeled data due to issues such as privacy concerns [13]. The aforementioned challenges motivate the implementation of machine learning (ML) techniques that are able to discover knowledge from data and make consistent predictions about human behavior [14]. Focusing solely on data-driven approaches that depend on large real-world datasets [15] is difficult because of limited data. On the other hand, taking a primarily knowledge-driven approach, which involves making numerous assumptions, tends not to be as robust because the assumptions might not hold across different smart homes [16].

HAR in smart homes is a problem that requires both segmentation and classification to be solved concurrently. Segmentation refers to identifying windows of interest (contiguous windows of sensor events relevant to an activity) before performing classification (i.e., recognizing activities associated with the identified window of interest). This dual problem is difficult to tackle for several reasons: (1) different activities have very different time windows (e.g., walking out of the home only takes 5 s, while sleeping can take as long as 10 h); (2) sparsity—most sensors are inactive in a typical day, (3) sensors have varying sampling rates—every home sensor potentially has a varied sampling rate. Consequently, defining an ideal window size is extremely challenging, leading to poorer performance, especially for HAR applications in smart homes.

Prior works have attempted to address this by (a) using fixed sliding windows (with limited success) or assuming the presence of an oracle (which is impractical) and/or (b) relying on alternative encoding techniques [17,18,19]. Unfortunately, HAR systems that rely on the assumption that an oracle can manually segment sensor streams are not very applicable to real-world scenarios because it is very time-consuming for residents to identify and filter segments in the deployment scenario [20]. Although a popular alternative in the literature is to rely on a fixed-length sliding window [21], we show that modeling discrete sensor events can outperform approaches with fixed-length sliding windows.

Building on the work by Ye at al. [21], we propose a novel graph neural network that is able to capture relationships between sensors by pooling node embeddings in a hierarchical manner. Since graphs are a natural way of representing networks, we posit that learning an expressive graph structure that models (1) the dynamics of sensor dependencies; (2) how those relationships evolve over time is key to addressing the main challenges associated with HAR. Our approach outperforms the existing state-of-the-art approaches on several publicly available smart home datasets. We also show how (1) it is more robust to intermittently faulty sensors; and (2) it can be a preliminary step in building more explainable HAR systems. Our compelling results across several scenarios and ablations strongly advocate for graph-based approaches to human activity recognition in smart homes.

The key contributions of this paper can be summarized as follows:We present a novel and more expressive graph-based activity recognition system for smart homes that better models inter-sensor relationships. This is built on top of a discussion on how to address an irregular sampling of sensor data streams, which are prevalent in smart homes.We provide a thorough experimental evaluation of challenging, real-world smart home problems which are encapsulated by a variety of CASAS smart home datasets.

## 2. Background

The primary task of HAR in smart homes consists of recognizing what a human is doing at what time, thereby analyzing sensor data that have been captured using a range of modalities, from the ‘Internet of Things (IoT)’ to environmental sensors [22], wearables, and cameras [23]. Given the trends of decreasing sensor costs and increasing demand for home automation, in this paper, we focus on IoT-based HAR.

### 2.1. Problem

IoT-based HAR is an inherently difficult problem, as it requires solving two sub-problems concurrently: segmentation (identifying contiguous–in time–sensor events that correspond to yet unknown activities), and classification (recognizing the resident’s actual activity covered by the previously identified segment of contiguous sensor events) [24]. Many HAR approaches, especially with wearables, rely primarily on the sliding window method, which is essentially a workaround that circumvents the explicit segmentation step. Specifically, contiguous sequences of sensor events in fixed time windows are treated as input to a HAR system. The window sizes are typically determined by guessing the window lengths that most likely capture a complete activity (e.g., a user wearing a smartwatch running) [25]. The time window is shifted along with some overlap. It is then typically, yet not always correctly [26], assumed that sliding windows are independent and identically distributed (i.i.d.).

Unlike wearables HAR, IoT-based HAR in smart homes is different along several dimensions. Primarily, the sensor data tend to be sparse (i.e., most sensors remain inactive for most of the day) and contain activities that span over a more varied range of durations. Consequently, no single window length can effectively capture all activities. Furthermore, unlike wearables, smart home sensors typically do not have a fixed sampling rate, making it more difficult to define an ideal window size. For these reasons, sliding window approaches do not work as well for many HAR applications in smart homes.

Prior works have attempted to improve the effectiveness of HAR in smart homes by (a) relying on an estimated fixed-length window (a workaround), (b) assuming access to an oracle that effectively guides the classification systems towards the portions of a sensor data stream that shall be classified, and/or (c) using different encoding techniques. We can group works into three broad categories: (1) traditional approaches, (2) general deep learning approaches, and (3) Graph Neural Network (GNN)-based methods. Traditional approaches tend to ignore temporal relationships of sensor activity. More recent deep learning approaches [17,18,19] tend to assume access to an oracle during deployment—the HAR system takes manually segmented windows of sensor events as input. Though feasible for developing and studying activity classification tasks, the assumption of having access to segmented sensor event streams is not valid for practical applications. Recent GNN methods [21], while promising, require fixed-length windows of sensor events which might not fully cover the varied durations of activities.

### 2.2. Traditional Approaches to HAR in Smart Homes

A plethora of (conventional) algorithms have been proposed for the task of sensor-based human activity recognition (HAR), ranging from naive Bayes (NB) and decision trees [27] to conditional random fields (CRF), hidden Markov models (HMM), and support vector machines (SVM) [28]. Clustering-based classification methods have also been proposed [29], where the k-nearest neighbors algorithm was used to determine resident activity. Beyond this, several variants of SVM have been shown to outperform traditional machine learning algorithms such as NB, CRF, and HMM [30]. Some authors have developed behavior classification models derived from SVM classifiers to differentiate/identify residents [31]. An SVM classifier using multiple kernels was also proposed to identify individual activities of residents [32]. Some authors focused on time-space feature importance relevance and used random forests and SVMs to distinguish relevant features for activity classification [33]. More complex probabilistic methods have also been proposed that leverage Bayesian networks [34], in addition to methods that estimate prior probabilities of activities happening at different points in time [35], relying on Gaussian mixture models. A disadvantage associated with many of these methods is that they typically require handcrafted feature extraction methods [36] or approximate kernel fusion methods [32]—an issue that is directly addressed by deep learning methods.

### 2.3. Deep Learning Approaches to HAR in Smart Homes

The key benefit of deep learning (DL) methods lies in their ability to uncover features from raw data (e.g., sensor measurements) [14]. Early successful DL works in the field of sensor-based human activity recognition proposed the use of a Restricted Boltzmann Machine (RBM) [22], which relied on a single-layer feed-forward network for feature extraction. Later works progress beyond a simple-feed forward network by suggesting the use of one-dimensional convolutional networks (CNNs) [37] and two-dimensional CNNs [38,39] that capture local dependencies in a temporal sequence. Local dependencies are captured through what is known as parameter sharing across time—the convolutional kernel used has the same weights across time. Consequently, CNNs have a restricted ability to capture dependencies between data samples [40].

LSTM networks have become more popular recently because of their ability to model long-term dependencies in sequences. In fact, they have been shown to be a very successful variant of recurrent neural networks (RNN) in automatically learning temporal information from raw data and achieving reasonable performance in HAR in smart home settings [17]. The authors also demonstrate the effectiveness of extensions of LSTMs such as bidirectional LSTMs and cascading LSTMs, attributing their success to explicit modeling of multi-modal channels of domotic sensors. Recent work has also shown the importance of good feature representations [41,42]. Bouchabou et al. [19] combine this insight with a popular natural language encoding technique, namely term frequency encoding, in generating good feature representations fed to a convolutional network. Although they seem to achieve state-of-the-art HAR results, many of these deep-learning methods have drawbacks: A key limitation is their reliance on oracle segmented sensor data, as outlined earlier. These segmented windows are often used as inputs to a classification model that identifies a specific human activity [17,19]. Yet, HAR systems that rely on manually segmented data are not very applicable to real-world scenarios because it is not practically possible to identify and filter segments that can be used for HAR in a deployment scenario.

### 2.4. Graph Neural Networks for HAR

In formulating the problem of HAR in smart home settings, most prior works implicitly ignore the heterogeneous structure of the data collected from a network of sensors. At any point in time, only a subset of all sensors will be activated, whereas other sensor measurements are not necessarily meaningful—which is in stark contrast to, for example, activity recognition scenarios using wearable sensors. Since different combinations of sensors can fire together at different points in time when a resident engages in an activity, sensor data associated with each resident activity are implicitly variable. Prior works ignore this heterogeneous structure. Apart from TLGAT [21], which attempts to construct a graph attention network across space and time, the state-of-the-art methods for HAR in smart homes are convolutional neural networks (CNNs) [18,19] and/or variants of recurrent neural networks (RNNs) [17], which do not factor in relational inductive biases and assume a fixed input size. In this context, relational inductive biases refer to assumptions that impose constraints on relationships and interactions between sensors. For example, when a resident of a smart home decides to watch television in the living room, their activity would probably trigger the living room light sensor and binary sensor, indicating if a couch is being used. In the context of this example, imposing relational inductive biases translates into encoding that the couch sensor and light sensor co-fired, and hence both of these factors are correlated (i.e., dependent on each other).

In real-world settings, where only limited labeled data are available and sensor data are highly variable, inductive biases are an excellent way to train models that generalize well [43]; better generalization leads to better adjustment to variations in resident behavior, leading to more robust activity recognition systems in smart homes. When working with heterogeneous data, relative inductive bias can be introduced through guiding models in learning dependencies between sensors. Several works in HAR use wearable sensor data and skeleton data that exemplify this: variants of spatial–temporal graph convolution network [44,45] and variants of residual graph convolutional networks [46,47]. While graph convolution networks have been successful in HAR for wearables, we show in this work that they tend not to be as successful as attention-based graph neural networks in HAR for smart homes, especially since attention-based models are able to leverage the prior knowledge, indicating that some neighbors might be more informative than others.

Knowing how a set of sensors are co-firing (e.g., sensors that typically trigger and do not trigger while a resident is watching the television) can provide more context when deciding the activity of a resident, even when only a subset of the sensors are functioning at a given point in time. Since graphs are an effective way to model dependencies in a network of sensors, one meaningful problem formulation involves defining a sensor network as a graph with nodes representing individual sensors and edges representing relationships between sensors. This leads to one of the key ideas pertaining to GNNs: we want to build representations of each sensor, which will comprise some learned combination of the sensor’s observed data (encoded as a node feature vector) and the feature vectors of correlated sensors.

Unlike prior works, we extend the work of Graph Neural Networks in the domain of human activity recognition in smart homes in several ways:Our approach does not rely on designing separate networks for individual features such as time, timestamp, and location. Our approach learns all the features automatically and directly from raw sensor observations. We believe this is helpful because by not splitting raw sensor data into separate features and training a unified graph neural network (Section 3) on less sparse data, learning is improved. This translates into better generalization, which is shown by the improved performance using our method.Secondly, our approach learns correlation relations between sensors across different features where we explicitly prune edges with low attention weights, allowing the learned graph structure to retain more pertinent information, allowing it to generalize better. More details in Section 3.4.1.Thirdly, we discontinue the use of convolutional filters, which were often found in prior works [21], by relying on the use of residual connections and hierarchical pooling (Section 3.5). Our experiments find that the use of convolution filters can be extremely sensitive to sensor behavior, window size, and chosen kernel size, potentially explaining the worse performance in methods using these filters (Section 4.2) compared to our approach.Another very important distinction lies in evaluation. Most prior works use a 70–30% split for training and test data with a random shuffle [18,21,36]. Unfortunately, this method is not representative of practical scenarios and is inappropriate for time series data because of temporal dependencies [37]. We show evaluations across several pertinent prior works using forward chaining. More details can be found in Section 4.1.2.

## 3. Graph-Guided Networks for Activity Recognition in Smart Homes

### 3.1. Overview

A successful HAR system in a smart home is one that is able to leverage a multitude of such sensors to effectively recognize user activities despite the challenges posed by real-world deployment. We argue that explicitly modeling relationships and interactions between sensors is a crucial step in addressing issues arising from variability, noise, redundancy, and sparsity in sensor measurements, especially when labeled user activity data are scarce. Knowing how different sets of sensors are co-firing can provide more context when determining the activity of a user even when sensors malfunction (e.g., due to poor network connectivity) or are activated for a short duration (e.g., a door sensor when the resident is leaving the home).

An elegant way to model relationships between an arbitrary number of entities is through graphs (cf. Section 2.4). We propose a novel graph-based method to model relationships between sensors and then leverage the graph to perform HAR in smart homes (see Figure 1 for an overview). Our approach consists of four main components:Encoding. What: A way to encode each sensor’s values into a vector that encodes both temporal information as well as sensor measurements into a vector. Why: Mapping observations to a high dimensional space using a non-linear transformation allows for sufficient expressive power [48].Sensor Embedding Generation. What: A sensor-specific transformation to capture the unique characteristics of each sensor. Why: Values recorded by each sensor can follow different distributions and have distinct patterns—hereafter referred to as ‘*sensor behavior*’.Attention-based Graph Structure Learning. What: A directed graph that represents the dependency relationships between sensors. It is learned through an attention function that attempts to quantify the strength of relationships between sensors.Why: Modeling dependency relations can provide more context for the task of human activity recognition. The graph is directed because dependency patterns need not necessarily be symmetric. Attention is needed since not all sensor dependencies are the same; some are perhaps more important compared to others.Activity Recognition. What: A modified fully connected neural network that uses hierarchically pooled information encoded in the learned graph to predict users’ activity. Why: The resultant graph from the previous step contains several dimensions of information: each node encodes temporal information and measurements from one sensor as well as weighted contributions from its neighbors; each edge encodes the strength and presence of dependence between sensors. By iteratively clustering and aggregating node embeddings using a fully connected neural network, the HAR system would be better able to piece all the dimensions of data encoded in the graph to make an informed prediction about a user’s activity.

In what follows, we provide detailed descriptions for each of the components of our approach.

### 3.2. Encoding

As outlined as one of the challenges in Section 1, variability in sensor measurements can manifest as signals that may be missing at random periods and for arbitrary durations. Several ways to address the issue of irregular sampling of sensor signals have been developed. Traditional imputation techniques, which are based primarily on averaging [49] or linear regression [50], do not necessarily encode the complexity that plagues smart home sensor signals. More robust approaches tend to be: (a) probabilistic, such as Gaussian Processes [51,52,53]; (b) kernel-based [54]; or (c) deep learning based, such as Generative Adversarial Networks (GANs) [55,56,57], Recurrent Neural Networks (RNN) [58,59,60], and attention mechanisms [61]. A major challenge associated with both probabilistic and kernel-based approaches for imputation is the complexity that arises from designing appropriate kernel functions or covariance functions for Gaussian processes in the multivariate case. In practice, because there are often limited labeled data when it comes to HAR in the context of smart homes, training GAN-like generative approaches to convergence is extremely challenging [62]. End-to-end deep learning approaches such as Neural Ordinary Differential Equation (ODE) networks [63,64] and Gated Recurrent Unit (GRU) networks with hidden states decayed towards zero [60] add too much model complexity, making the HAR system more prone to overfitting to a specific resident or smart home environment.

We found that a straightforward solution that adds minimal complexity and works well is forward imputation, i.e., assuming any missing value is the same as its last measurement and thus using the most recent sensor measurement. Hence, the main data transformation we perform for raw sensor events is to perform forward imputation (the first step in Figure 1). Discrete sensor events are then fed into an encoder that is able to capture contextual information across time. The encoder converts $x_{i}$ to $h_{i}$ as shown in the second step in Figure 1. Since variants of Recurrent Neural Networks (RNN) such as LSTM and Bidirectional LSTMs are known to be effective in capturing long-term dependencies, we chose to use an RNN-based encoder to extract feature vectors, $\left\{h_{1},\ldots h_{N}\right\}$, from the raw sensor data, $\left\{x_{1},\ldots x_{N}\right\}$, as shown in step 2 of Figure 1.

### 3.3. Embedding Generation

Across various smart homes, different sensors might have different behaviors and characteristics along several dimensions: (i) frequency of activation, e.g., sensors placed along hallways might trigger more frequently compared to sensors at the door for a resident who mostly stays at home; (ii) periodicity of triggering, e.g., bedroom sensors might be activated daily at night while the guest bathroom sensors might only activate during the holiday season when a guest visits; (iii) range of measured values—discrete-value sensors such as binary sensors with two values (On and Off) versus continuous-value sensors such as temperature sensors and air-quality sensors, which can take a wide range of values. We refer to variations along these dimensions as ‘*sensor behavior*’.

In order to capture the aforementioned dimensions of variations in ‘*sensor behavior*’, our approach includes an embedding layer with a size *N*, which represents the number of sensors. Each sensor’s embedding, $s_{i}$, is defined as follows:(1)$$\begin{array}{cc} s_{i} & =ReLU(h_{i}\cdot W_{i}) \end{array}$$(2)$$\begin{array}{cc} s_{i} & \in R^{d},\mathrm{for}\;i\in \left\{1,2,3,\ldots N\right\} \end{array}$$
where ReLU is a non-linearity function, $W_{i}$ is a sensor-specific trainable weight vector, and *d* refers to the dimension of the embedding. Together, the non-linear transformation projects feature vector $h_{i}\in \left\{h_{1},\ldots h_{N}\right\}$ from the encoding step (Section 3.2) to a new feature space. In this new feature space, sensors that have similar *‘sensor behaviors’* share similar embeddings, indicating a high tendency to be related to one another. The learned embeddings are used to construct the graph structure, which maps the relationship between sensors. They are also used to compute attention scores, which determine how much each sensor affects the values of related sensors. This provides a way to address variability in sensor measurements, e.g., when sensors malfunction. Initialized randomly, the sensor embeddings, which are essentially semantic clusters, are learned alongside the rest of the model.

### 3.4. Attention-Based Graph Structure Learning

Knowing how a set of sensors are co-firing can provide more information when determining the activity of a user in real-world settings. By explicitly modeling relationships and interactions between sensors using graphs and then leveraging the graphs to recognize human activities, we believe that HAR systems would be better equipped in addressing issues arising from variability, noise, redundancy, and sparsity. For example, when sensors malfunction due to poor network connectivity or are activated for an extremely short duration (e.g., a door sensor activating only when a user leaves home), being able to rely on information from other sensors would allow a HAR system to have sufficient context to predict user activity. Concretely, we refer to the process of modeling relationships between sensors using graphs as attention-based graph structure learning.

A graph is a collection of nodes and edges. In the context of our proposed HAR model, the graph comprises nodes that represent features associated with specific sensors and edges that indicate relations between sensors. A directed edge $\vec{e_{uv}}$ from a sensor *u* to another sensor *v* can be interpreted as sensor *u* potentially influencing the behavior of sensor *v*. Since the dependency between *u* and *v* might not be symmetric, we explicitly model a directed graph implemented as an adjacency matrix *A*, where $A_{uv}\in \left\{0,1\right\}$ represents the presence of a relationship between sensor *u* and sensor *v*: when sensor *u* captures an observation, sensor *v* will receive a neural message. If there is no edge connecting sensor *u* and sensor *v*, there is no exchange of neural information between both sensors, indicating that the sensors are unrelated. As detailed in Section 2.4, neural message passing is equivalent to incorporating the hidden state (i.e., feature vectors) of any given sensor *u* into the feature vectors of all potentially dependent (i.e., co-firing) sensors using a weighted aggregation function. Serving as an illustrative example, consider the CASAS Milan smart home that has a regular layout. Each sensor (e.g., M07, M08, and M26) in the house (Figure 2) has a corresponding node in the subgraph on the right representing its feature vector. The presence of directed edges, e.g., between M07 and M26, indicates that M07 influences M26, and vice versa. Consequently, the neural message-passing algorithm incorporates the features of M07 into the the feature vector of M26.

#### 3.4.1. Edge Pruning

Since it is reasonable to assume that not every sensor will influence every other sensor, the dependency relations between sensors need to be determined. Unlike prior works [21], we include an explicit edge pruning step where at each training step we prune edges between nodes with small attention weights, allowing the learned graph to primarily focus on sensor relations that are actually pertinent for the task of activity recognition. We believe this directly addresses the high level of sparsity present in smart home sensor observations. Starting with a fully connected graph where every sensor *i* has directed edges going to and coming from all other sensors, we can remove unnecessary connections—in other words, relations that do not necessarily provide additional information. Note that we can also start with a partially connected graph if we have prior information about the sensor relationships. The importance of edges can be determined by comparing the similarity of sensor embeddings, which were designed to encode sensor behaviors.

More formally, we use the normalized dot product of sensor embeddings generated from the embedding layer, which is computed between every pair of candidate relations (step 2 in Figure 3) as follows:
(3)$${\mathrm{sim}}_{ij}=\frac{s_{i}^{⊺}s_{j}}{∥s_{i}∥\cdot ∥s_{j}∥}\;\mathrm{for}\;j\in C_{i}$$
where $s_{i}$ is the embedding of sensor *i* and $C_{i}$ is a set of all potentially related sensors to *i* (i.e., sensors that co-fire). For each sensor *i*, we sort and only keep edges corresponding to the top *k* similarity scores, where *k* is a hyperparameter (steps 3 and 4 in Figure 3), which we have chosen to be five through empirical study. Discarding non-top-k edges translates into updating the adjacency matrix *A* of the graph.

#### 3.4.2. Attention Mechanism

Some sensor pairs can be more closely related (i.e., correlated) to each other than others. For instance, sensors associated with a ceiling light and fan in the living room are probably more likely to co-fire when a resident is sitting and watching television in the living room. Compare this to a ‘weaker’ dependence between sensors associated with a bed lamp in the bedroom and hallway. A resident walking through the hallway could be on their way to carry out any activity: to the bedroom to read, to the living room to watch television, or to the kitchen to cook, to name but a few examples. Clearly, it is reasonable to expect the dependence between light and fan sensors in the living room to be stronger than the dependence between sensors associated with the hallway and bed lamp in the bedroom. In order to effectively learn such nuanced sensor relations, we include attention-based [66] weights for every directed edge $\vec{e_{uv}}$ between sensors *u* and *v* where {u,v}∈S and *S* is the set of all sensors in a given smart home.

More specifically, a graph attention-based feature extractor performs a weighted aggregation of feature vectors from related neighbors to update each node’s representation $z_{i}$, i.e., sensor *i*’s hidden state. It is defined as follows:(4)$$\begin{array}{cc} z_{i}^{\left(t\right)} & =\mathrm{ReLU}\left(\beta_{i,i}Wx_{i}^{\left(t\right)}+\sum_{j\in N(i)}\beta_{i,j}Wx_{j}^{\left(t\right)}\right), \end{array}$$
where $x_{i}^{\left(t\right)}\in R^{N\times v}$ and *N* is the number of sensors, and *v* is the window size on input sensor readings. N(i) is the set of neighboring sensors of (i.e., sensors related to) sensor *i*, obtained from the adjacency matrix *A*. *W* is a trainable weight matrix, which is used to apply a shared linear transformation to the model input $x_{i}$. $\beta_{i,j}$ are the normalized (using softmax) attention weights, which are computed using the following equations:(5)$$\begin{array}{cc} \beta_{i,j} & =\frac{\exp\left(\phi \left(i,j\right)\right)}{\sum_{k\in N(i)\cup \{i\}}\exp\left(\phi \left(i,k\right)\right)} \end{array}$$(6)$$\begin{array}{cc} \phi \left(i,j\right) & =\mathrm{LeakyReLU}\left(b^{⊤}\left(g_{i}^{\left(t\right)}⊕g_{j}^{\left(t\right)}\right)\right) \end{array}$$(7)$$\begin{array}{cc} g_{i}^{\left(t\right)} & =s_{i}⊕Wx_{i}^{\left(t\right)} \end{array}$$
where $g_{i}^{\left(t\right)}$ is the concatenation of sensor *i*’s embeddings with the corresponding transformed feature vector of sensor *i* and *b* is a set of learned coefficients. Note that our approach specifically factors each sensor’s behavior encoded in the embeddings, $s_{i}$, unlike prior graph attention mechanisms.

The attention mechanism in conjunction with the edge-pruning strategy allows us to learn a directed graph that encodes both sensor-specific features as well as inter-sensor relationships. In the next subsection, we outline how this rich representation can be used to recognize the activity a user is engaged in.

### 3.5. Activity Recognition

An effective HAR system that learns dependencies between sensors ought to answer several questions:Are sensors co-firing in patterns we have observed before?Which sensors have been active?What can we learn about user activities by looking at sensors that are active as well as those that are not?

These questions can be answered by querying the learned graph structure. More specifically, we apply a function $f_{pool}$, which pools the information encoded in the graph to predict user activity. This function ought to meaningfully map the set of node feature vectors $\left\{z_{1},\ldots z_{N}\right\}$ in a learned directed graph to a specific user activity (Step 5 in Figure 1). A natural implementation of $f_{pool}$ would be to use a single fully connected layer, since they are known to be universal function approximators [67]. Unlike prior works, we modify the standard fully connected layer using a differentiable soft assignment of the graph by mapping nodes to sets of clusters based on their learned embeddings (in a method similar to that described in [68]). Essentially, this translates into nodes being pooled iteratively to aggregate both local and global graph embeddings. The aggregated embeddings are then used to learn to map features from sensors and information encoded in the directed graph to specific user activities. Also, unlike prior works, each of the *N* representations $\left\{z_{1}^{\left(t\right)},\cdots ,z_{N}^{\left(t\right)}\right\}$ are element-wise multiplied with the corresponding sensor embedding $s_{i}$ and fed into the fully connected layer of a size equivalent to the total number of desired activities for classification. This is equivalent to having a skip connection, minimizing the effects of over-smoothing [69]. Consequently, we have a complete HAR system that takes as input discrete sensor measurements $\left\{x_{1},\ldots x_{N}\right\}$ (data preprocessing is outlined in Section 4.1.1) and predicts user activity for that duration. In practice, our system can be used as is without the need for any oracle manually segmenting windows or for having to guess the ideal window size of input window sequences.

## 4. Experimental Evaluation

In this section, we report on our extensive and rigorous experimental evaluation, which aims to test the effectiveness of our proposed approach in plausible, real-world scenarios, and compare our approach to state-of-the-art methods. The natural choice for datasets to be explored in a rigorous experimental evaluation is the CASAS datasets [65]. They contain diverse collections of daily activities data collected in both single and multi-person households with real users, thereby covering time spans between two and eight months and covering a range of demographics, from young adults to older adults with dementia, and even pets. The houses from which data were collected featured a varied set of sensors, including temperature sensors, motion sensors, and binary sensors such as door sensors (which capture whether or not a door was opened). Together, the diversity of residents (and their behaviors) as well as the variety of sensors used make the CASAS datasets reflective of real-world settings.

The subset of CASAS datasets we utilize for evaluations portray the previously mentioned four main challenges that are pervasive in real-world settings: sensor variability, sparsity in measurements, the presence of noise, and limited availability of activity data. In fact, they even contain months of labeled activities that are severely imbalanced, i.e., certain activities occur way more frequently than others, rendering the problem more challenging. What follows are evaluations performed using several CASAS datasets from Aruba, Cairo, Kyoto7, Kyoto8, and Milan (Table 1).

### 4.1. Evaluation Setup

#### 4.1.1. Data Preprocessing

For each of the datasets, we follow a process of cleaning up, which is similar to the process described by Bouchabou et al. [18]. Specifically, we address the following major anomalies found in each of the raw datasets: (1) the presence of duplicate data—for example, sequences of sensor activations for some days are repeated—and (2) incorrect ordering of sensor activations, i.e., some sensor activations do not appear chronologically. Duplicate data are removed and incorrectly ordered sensor activations are reordered to be chronologically consistent. Furthermore, since the raw data only contain the “start” and “end” of activity labels at specific timestamps, we process raw data into steps, where a step for a dataset is defined as the smallest interval in time between consecutive sensor events. For steps during which no observation data are available, we use forward imputation, i.e., we use the last known observation data. More details are provided in the Appendix B.

#### 4.1.2. Evaluation Methodology

Unlike prior works [17,18,19], we are unable to use the stratified K-fold cross-validation method [70] since the adjacent segments of observed data are not statistically independent. Relying on stratified K-fold cross-validation results in bias towards approaches that preserve similarity between adjacent segments of time-series data, affecting the generalizability of classifiers [26]. Hence, we use a more principled procedure: forward chaining [71,72]. We split each dataset into three consecutive segments for three-fold evaluation, as illustrated in Figure 4. In each fold, a subsequence of sensor events is used in the training process, which is called the training subsequence (highlighted in yellow in Figure 4). The training subsequence is further split into a training subset (highlighted in purple in Figure 4) and a validation subset, which comprises 10% of the training subsequence (highlighted in green in Figure 4). In the same fold, an adjacent unseen subsequence is identified for testing (highlighted in red in Figure 4). In the next fold, the training and testing subsequences from a previous fold are used for training in the next fold, and an unseen subsequence of sensor events is used for testing. This process is repeated three times to obtain three different F1 scores that cover all samples. The reported scores are the average of all three folds of evaluation over several runs.

We compare validation and training losses to determine if training should terminate early, i.e., before overfitting. This is achieved through early stopping [73], where training is terminated if the validation loss does not decrease for *S* epochs; we chose *S* to be 15. Setting a larger value for *S* led to models overfitting. With early stopping, all models, even those set to train for 300 epochs, terminated early, at between 50 and 80 epochs, depending on the dataset. The learning rate is set to 0.001 using the Adam optimizer with a weight decay of 0.5%.

Model hyperparameters. We have used a batch size of 128. The number of nodes in the graph is dependent on the number of sensors in the dataset (Refer to Table 1). We start with fully connected graphs and retain the top five edges in each step of edge pruning (refer to Section 3.4.1). In order to ensure a fair and consistent comparison with other state-of-the-art methods, we use a window size of 20 with a 50% overlap. A window size of 20 in our case means that in each window there are 20 timesteps, where the duration of each timestep is determined to be the shortest interval for which any sensor has observations.

### 4.2. Results for Different Evaluation Settings

#### 4.2.1. Setting 1: Classic Time Series Classification

The first experiment evaluates how well the proposed approach is able to identify resident activities using sensor measurements across time windows. Each of the chosen datasets is pre-processed as outlined in Section 4.1.1.

Our graph-guided approach outperforms prior works across several CASAS datasets. We compare our approach to several other approaches: (1) Common recurrent baselines—Long Short-Term Memory (LSTM) and One-dimensional Convolutional Neural Network (CNN), (2) recent Graph Neural Network approaches used in HAR for wearable devices, (3) language embedding-based methods single-resident daily activity recognition in smart homes (4) several other benchmarks for single-resident daily activity recognition in smart homes. Note: Several of the baselines had to be modified to be suitable for comparison (e.g., A3TGCN [74] was designed for forecasting but had to be modified to be made suitable for classification—the change loss function was altered and the forecasting head was replaced with a classification layer. There are some related works such as that of Cook et al. [30], who evaluated only private datasets to which we do not have access; hence, we cannot compare our results with these works.

We have had several key findings. Firstly, several prior works that focused on deep learning methods relying on natural language processing (NLP) techniques [18,19], though seemingly effective, required an oracle to segment windows of sensor events. Fundamentally, these methods require external segmentation information in the form of text labels (e.g., ON, START, etc.) during deployment. It is unrealistic and impractical to expect residents to label time windows before performing activity recognition. Secondly, graph-based methods typically tend to outperform common recurrent approaches such as Convolutional Neural Networks and LSTMs, as can be seen from the F1 scores in the first two sections of Table 2.

Thirdly, graph-based methods that rely on graph convolutions tend to perform worse than graph-based methods that use self-attention at their core. This is evident from the significant improvement of at least ≈10% in F1 scores (Table 2) on CASAS Kyoto8, Milan, and Kyoto7 by using Graph Attention Networks (TLGAT [21] and our approach) as opposed to Graph Convolutional Networks such as A3TGCN [74], ST-GCN [77] and, GraphConvLSTM [45]. Unlike TLGAT [21], our approach takes the benefits of a graph attention network a step further in several ways: 1. Our approach combines LSTM encoders with a single unified graph instead of performing 1D convolutions over several different networks. 2. Our approach explicitly relies on learned attention weights at each training iteration to prune edges that do not strongly represent the correlation relationship between any pair of sensors. 3. We propose the use of residual connections (connecting sensor embeddings directly to graph embeddings) instead of using a set of convolution layers to aggregate node embeddings. We attribute the improvement in performance to architecture modifications that allowed our network to minimize the effects of over-smoothing [69]. Over-smoothing occurs when node-specific information is lost after iterations of message passing (i.e., node representations converge to indistinguishable vectors). Incorporating skip connections between sensor encodings (before and after message passing), our GNN approach does not seem to suffer from over-smoothing. Furthermore, the hierarchical pooling of node embeddings allows for learning multiple relationships that leverage the graph structure. Unlike prior works, we do not just concatenate all node embeddings randomly; we perform soft clustering of node embeddings iteratively, allowing for the co-firing relationships to persist after pooling. Consequently, our approach is able to achieve significantly higher than state-of-the-art F1 scores across several CASAS datasets.

Fourthly, the gain in performance by using our approach as measured by the F1 score is largest in Kyoto8 and lowest in Cairo. This phenomenon can be explained in part by the difficulty of the datasets. There are approximately four times more sensor activations in the Aruba dataset as opposed to the Milan dataset. Having limited collected data exacerbates the difficulty of HAR, since fewer examples are available to effectively learn user behavior. The fact that our approach significantly outperforms the state-of-the-art method on both more challenging datasets such as Milan and Kyoto8, as well as on less challenging CASAS datasets, is a testament to the robustness of our proposed approach.

#### 4.2.2. Setting 2: Fixed Sensors Dropped

In smart homes, it is reasonable to expect missing observation data because sensors can suddenly stop working—perhaps because they malfunction when exposed to external forces or their battery runs out. We posit that learning dependencies between sensors can be extremely beneficial in such scenarios. To this end, we evaluate if our approach is robust to scenarios where observation data from a subset of sensors is not available. One way to test this would be to permanently hide all measurements from the most informative sensors in the test data, keeping the training data unchanged. Training data are (still) required to learn dependencies between sensors. Informativeness is determined by how often each sensor is activated.

As shown in Table 3 (under ‘Set. 2 F1’ columns), our approach (evaluated on three CASAS datasets) is very effective. For instance, for the Milan and Kyoto7 datasets, our approach achieves F1 scores of 53.8 and 79.9, even when observation data from 40% of the most informative sensors are missing. Another recent graph-based baseline [21] also remains robust to sensor dropping, which makes the case for graph-based methods, although it still does not perform as well as our approach. In comparison, the RNN-based state-of-the-art approach collapses to an F1 score of only 8.00 and 35.1 correspondingly. Another finding is that the drop in effectiveness is not as significant (less than 10%) for our approach (and [21]) when the percentage of missing sensor data is 10%, 20%, or 30%) across all three datasets. On the other hand, the decrease in the effectiveness of the prior state-of-the-art approach is much larger. For instance, when just 10% of sensor data are missing (Milan), the F1 score of Deep CASAS BiLSTM drops from 45.5 to 31.7, while that of graph-based approaches like ours only drops by less than five points, from 80.4 to 75.5.

#### 4.2.3. Setting 3: Random Sensors Dropped

Instead of only hiding observation data from the most informative sensors, as described in Setting 2 (Section 4.2.2), it is also possible for different subsets of sensors to malfunction over time. We test this by hiding the observation data of randomly selected subsets of sensors for each window in the test set, again with several CASAS datasets. The results are reported under ‘Set. 3 F1’ of Table 3. The evaluation for each missing sensor ratio is repeated 10 times to prevent selection bias from strongly influencing the results.

We found that, generally, the effectiveness of both methods was better in Setting 3 than in Setting 2 because in the former it was still possible for data from the more informative sensors to help with activity recognition. In the latter, we systematically removed data from the most informative sensors. Similar to our finding in Section 4.2.2, our approach continues to be robust even when we randomly remove observation data from up to 40% of the sensors. In contrast, the effectiveness of non-graph-based approaches such as BiLSTM drops steeply. In both scenarios, our approach still outperforms or at least performs as well as other smart home HAR baselines.

Through both evaluation scenarios (Settings 2 and 3), we show that our graph-based approach is more robust to different degrees of missing sensor data as opposed to the state of the art. One takeaway from this is that graph-based approaches can be more useful than RNN-based methods in real-world deployments where sensor data can be missing for several reasons, including (but not limited to) frequent power cuts and external forces of nature.

### 4.3. Ablation Study

Considering that our approach contains several components, as outlined in Section 3.1, we investigated the contributions of the different components to the effectiveness of HAR. Table 4 summarizes our findings, which were obtained by conducting several ablations. For example, the graph structure seems to contribute the most towards being effective in recognizing user activity (measured by a ≈30% improvement in the F1 score). Although the inclusion of an embedding layer seems to only improve the F1 score by approximately 2% across several datasets (comparing Graph and Graph + Embedding in Table 4), the inclusion of the attention mechanism seems to improve F1 scores by at least ≈5%. Table 4 shows that all components are necessary and that the proposed graph structure helped improve the effectiveness of our proposed approach across several diverse CASAS datasets.

## 5. Discussion

In designing and evaluating the graph-guided neural network presented in Section 3, we have gained several insights with respect to design decisions and what future research directions could entail. For brevity, we only discuss some of these insights that contextualize our work. We have deferred additional discussion to the Appendix C.

### 5.1. What Are Some Limitations & Future Work?

Evaluation. As detailed in Section 4.1.2, our evaluation methodology uses nested forward-chaining. Specifically, we create several train/test splits and average the results across all the splits in order to reduce bias in evaluation and factor temporal dependencies in sensor events. However, in our evaluation setup, there is a possibility for bias arising from delays (in days) between train and test splits. We minimize the bias by repeating the evaluation with different values of K in K-fold and averaging it.

GNN architecture limitation(s). The message-passing framework (detailed in Section 2.4) is a crucial component in learning the graph structure (Step 4 of Figure 1). One of the unintended effects of message passing is over-smoothing [69], where a single node’s representation becomes dominated by too much information from all of its neighboring nodes. In the context of a sensor network, this is equivalent to irrelevant sensors corrupting the representation of any single sensor. As a corollary, the representations encoded by each node in the graph are less useful, leading to poorer accuracy in recognizing resident activity. One solution is to include skip connections to minimize the loss of information, as detailed in Section 3.5. Another intuitive solution we incorporate, unlike prior works, is to constrain information between a node and its top-K most important neighbors, where importance is determined using attention scores (or attention weights), as detailed in Section 3.4.2.

Figure 5 shows how the effectiveness of the HAR system varies with the number of neighbors each sensor receives neural messages from. We noticed that empirically choosing k to be between four and six yields the best performance across several real-world datasets. For the rest of the values of K, we observe a non-monotonic trend—any value of K less than five is insufficient information, and above that, there is too much information flowing into any specific node during message passing. This directly translates into concave curves centered approximately where K equals five, a pattern which has been observed in several CASAS datasets (Figure 5). A potentially meaningful direction for future work would be to look into whether we can overcome the limitation of choosing a fixed value for K.

Generalizability. Theoretically, our proposed graph approach can be easily used to bootstrap homes with similar layouts. However, since transfer learning is out of the scope of this work, we do not conduct any studies where we test how well learned graph structures can be used in new home layouts. We leave this for future work.

### 5.2. How Scalable Is the Proposed Approach?

One of the main strengths of our approach is its scalability. Since every node in the graph represents a sensor’s embedding and every edge is the co-firing relationship between a pair of sensors, removing sensors in a smart home would be equivalent to removing nodes in the learned graph representation. From experiments in Section 4.2.2 and Section 4.2.3, it is evident that our graph-based HAR system is pretty robust to both random and specific sensors being removed, without any retraining. In scenarios where only 1–2 random sensors are removed, we observed that, with retraining, we can improve the F1 score to within 2% of the setup where no sensors are removed. The caveat is that if a specific sensor that is removed is important for a specific activity, we observe a sharp drop in the effectiveness of activity recognition. For instance, removing the bedroom sensors altogether resulted in the HAR system barely being able to recognize the resident activity of ‘sleep’ on the CASAS Milan dataset.

In the case where new sensors need to be added to the system, it is also easy to add them without having to retrain everything from scratch; only finetuning is needed to refine the edge connections of the new sensors to all existing sensors in the learned graph representation. No modifications to the training paradigm are necessary. If we know that a newly added sensor behaves similarly to an existing sensor, we can bootstrap the new sensor’s embedding using the similar sensor’s embedding. We can further improve the effectiveness of the HAR system by finetuning the graph structure with the existing attention-based graph structure learning paradigm by starting with a partially trained graph. In future work, we aim to create datasets with realistic removal and addition of sensors, using which we can robustly quantify the scalability of our and other similar works in the context of HAR in smart homes.

### 5.3. Can We Explain the Classifications Performed by Our Proposed HAR System?

Most activity recognition systems are not perfectly accurate. Current HAR systems are not able to recognize every activity of every resident in households without any error. Although it is reasonable to expect activity recognition systems to improve, expecting them to be perfectly accurate seems unrealistic, specifically in the smart home setting, where there is a lot of diversity in terms of occupants and their behaviors. Unexplained, inconsistent behavior of activity recognition systems can give rise to smart home operations that tend to be surprising or even inappropriate to the resident. Given this context, we investigate the potential for explainability that our proposed approach might carry.

The graph-guided model inherently allows for more explainable classifications because we are explicitly modeling both the presence of a relationship between any two sensors in a sensor network and the strength of such a relationship using attention. The strength of relationships between sensors, as measured by the attention score, can be very informative in explaining classifications. For instance, in accurately recognizing the activity of sleeping for the resident from the Milan dataset, we can identify which sensors were deemed important by the model. To achieve this, we rely on attention scores (or attention weights), which are similarity scores that are computed by taking the normalized dot product of observations for all pairs of input sensor events as well as between any two sensors’ feature vectors. If we were to plot the normalized attention scores (a proxy for relationship strength) between all pairs of sensors in Milan, we would obtain an attention map as shown in Figure 6a. It is evident that sensor 28 (M028) has the highest attention score.

Looking at the annotated layout of the household used in Milan (Figure 6b), we can see that sensor M028 is located in the master bedroom. From this, we could infer that for an input window of sensor measurements, the model predicts *sleep* as the resident’s activity based on the high sensor activity of the sensor in the master bedroom.

Explanations have also been useful in determining the need for curriculum learning. When analyzing incorrect activity predictions, we found that they often corresponded to high attention weights associated with incorrect sensors. For example, the activity of *leaving home* was often incorrectly identified and the sensors deemed most pertinent were the hallway sensors (M011 and M019). What makes this outcome unsurprising is that not only were the wrong sensors identified but the corresponding activity was one that occurred more frequently, such as doing *chores*. We used this insight to structure the training process such that we reintroduce difficult training samples–training samples that more intensely demonstrated the issue of noise and redundancy–as training evolved. We hope that such explanations will not only help improve training processes but that they will also instill more trust in HAR systems, especially in extenuating circumstances where such a system is bound to fail.

## 6. Conclusions

A growing number of applications have focused on Human Activity Recognition (HAR) in smart homes, specifically in the field of ambient intelligence and assisted-living technologies. Real-world deployments of HAR systems pose several challenges, such as variability, sparsity, and noise in sensor measurements. The complexity of the aforementioned challenges motivates the implementation of machine learning (ML) techniques that are able to discover knowledge from data and recognize human activities. Although deemed effective in recognizing resident activities, most recent works assume that during deployment, smart home residents are able to segment sequences of discrete sensor events before the HAR system is able to perform activity recognition. We call this the oracle-guided segmentation problem. Unfortunately, relying on such data is not very applicable to real-world scenarios, since it is not practical to expect residents to identify and filter sequences of sensor data. Similarly, relying on fixed window lengths is another common limitation of prior works.

In an attempt to address the aforementioned issues, we have developed a graph-guided neural network for HAR, thereby avoiding the previously mentioned limitations. The key insights leveraged in our work are as follows: (1) explicitly learning dependencies between sensors in a hierarchical manner can make for a robust HAR system; (2) discrete sensor events can be effectively used for smart home HAR without the need for oracle-specified segmentation or fixed window lengths. In our evaluations, we demonstrated the effectiveness of our proposed HAR system in several settings. We also showed how our proposed approach allows for more explainable classifications. Ultimately, we hope that graph-based approaches will become integral in designing HAR systems in real-world deployments of smart homes.

## Figures and Tables

**Figure 1 sensors-24-03944-f001:**
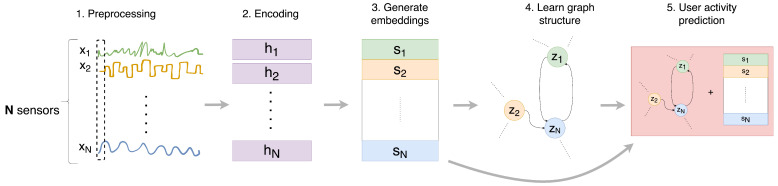
Overview of the proposed GNN-based approach to human activity recognition in smart homes. Inputs $\left\{x_{1},\ldots x_{N}\right\}$ correspond to each of the *N* sensor observations. (1) Preprocessing: Forward imputation is performed if necessary. (2) Encoding: In the encoding step, an encoder applies non-linear transformations to inputs $\left\{x_{1},\ldots x_{N}\right\}$ to generate representation vectors, $\left\{h_{1},\ldots h_{N}\right\}$. (3) Sensor Embedding Generation: The encoded inputs are then used to generate sensor-specific embeddings $\left\{s_{1},\ldots s_{N}\right\}$. (4) Attention-based Graph Structure Learning: The sensor-specific embeddings are used to learn dependency relations, i.e., the edges between nodes. (5) Activity Recognition: All sensor embeddings are combined with the learned graph structure {*z*_1_,… *z_N_*} using a modified one-layer feed-forward neural network to predict user activity (we perform a hierarchical pooling of node embeddings to maximize the learning of global and local sensor relations).

**Figure 2 sensors-24-03944-f002:**
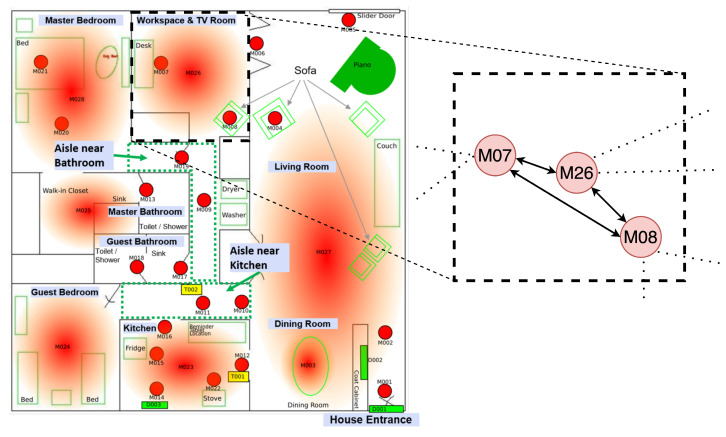
Layout of CASAS Milan smart home with labels (with permission from [65]) with the corresponding subgraph for the workspace and television room. The remaining nodes and edges of the graph are hidden for brevity.

**Figure 3 sensors-24-03944-f003:**
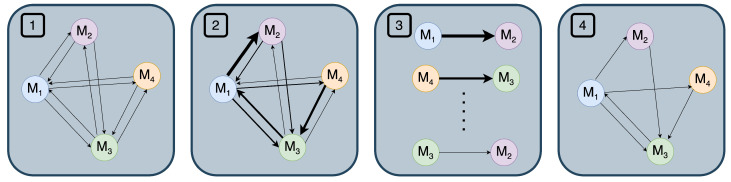
An overview of how we perform edge pruning on a 4-node sample graph. (1) A fully connected or partially connected graph is constructed. (2) $sim_{ij}$ between sensors *i* and *j* are computed. (3) The similarity scores of all sensors are sorted in descending order. (4) Only the edges corresponding to top-k similarity scores are retained, while the rest are discarded.

**Figure 4 sensors-24-03944-f004:**
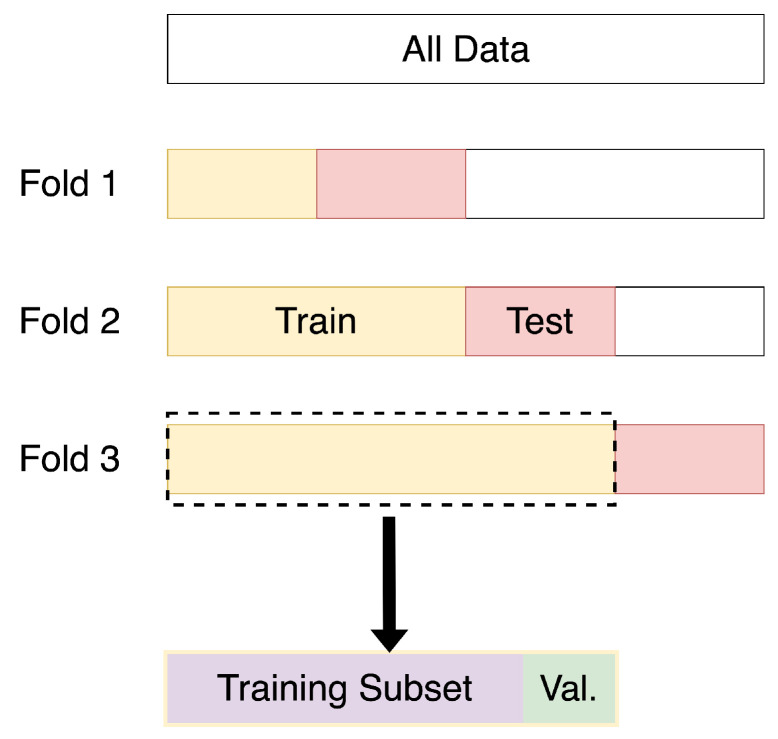
Overview of evaluation methodology involving forward-chaining for a three-fold evaluation.

**Figure 5 sensors-24-03944-f005:**
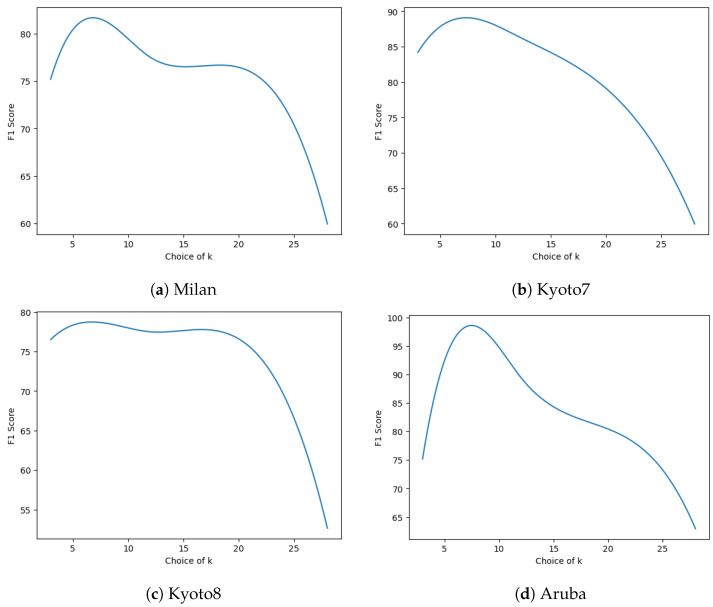
Plots comparing how the F1-score varies with the number of neighbors K chosen to aggregate information. The subfigures above show how varying the number of neighbors for each sensor affects the effectiveness of the HAR system. At each step of AGGREGATE in the neural message passing framework (Section 2.4), if a sensor has too many or too few neighbors, the graph is not able to learn a good representation, which reduces the effectiveness of the HAR system. Based on empirical evidence, we make a heuristic decision and select the choice of K to be five, which happens to generalize well across all datasets considered in our evaluations.

**Figure 6 sensors-24-03944-f006:**
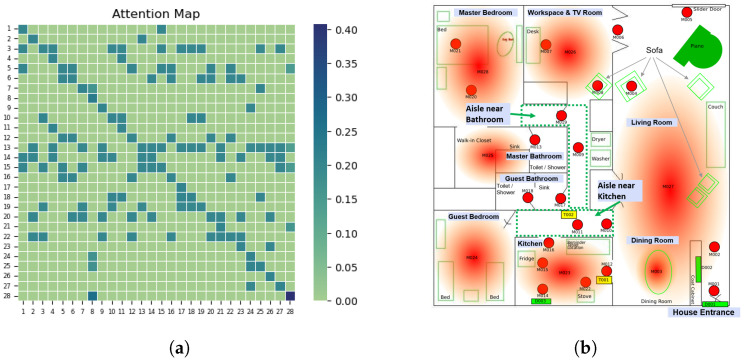
(**a**) Attention map, (**b**) Milan layout (with permission from [65]). Explanation for one occurrence of sleeping activity of a resident from the Milan dataset. The attention map on the left (Figure 6a) shows that sensor 28 (M028) has the highest attention score, with the darkest cell on the map. Looking at the Milan layout on the right (Figure 6b), M028 is the sensor identifying if a resident is on the bed in the Master Bedroom. Based on this information, we can infer the following: Since the resident is on the bed, the HAR system recognizes that the resident is sleeping. Being able to attribute recognized activities to its associated sensors through attention can help shed some light on the decision-making process of the HAR system, which, in turn, allows trust to be built with such systems.

**Table 1 sensors-24-03944-t001:** Details of CASAS datasets used for experimental evaluation.

Properties	Aruba	Cairo	Kyoto7	Kyoto8	Milan
Residents	1	2 + pet	2	2	1 + pet
Number of sensors	39	27	58	61	33
Number of activities	12	13	13	12	16
Number of days	219	56	46	58	82

**Table 2 sensors-24-03944-t002:** Classification performance on multiple CASAS datasets, on a per-sample basis. Comparisons are made across five categories of works: (1) Common recurrent baselines—Long Short-Term Memory (LSTM) and One-dimensional Convolutional Neural Network (CNN)—(2) recent Graph Neural Network approaches used in HAR for wearable devices, (3) language embedding based methods for single-resident daily activity recognition in smart homes (4) all other benchmarks for single-resident daily activity recognition in smart homes, and (5) our proposed approach, which does not require fixed windows or oracle specified time windows. We report the mean and standard deviation of the three-fold test F1-score across three runs.

Methods\Datasets	Kyoto8	Milan	Kyoto7	Aruba	Cairo
	F1 Score	F1 Score	F1 Score	F1 Score	F1 Score
CNN1D	26.6 ± 0.66	36.6 ± 0.36	74.9 ± 0.29	65.6 ± 0.43	80.5 ± 1.67
LSTM	22.7 ± 1.95	30.6 ± 0.85	76.4 ± 0.88	83.7 ± 0.26	81.1 ± 1.07
TCN-AE [75]	26.2 ± 1.22	43.5 ± 2.32	69.4 ± 2.84	85.5 ± 1.01	80.44 ± 1.75
LSTM-CNN [76]	24.9 ± 0.22	40.1 ± 0.39	81.5 ± 0.72	88.9 ± 0.40	83.95 ± 0.48
GraphConvLSTM [45]	30.1 ± 1.56	41.0 ± 1.98	77.3 ± 0.73	87.6 ± 0.12	84.2 ± 0.11
ResGCNN [46]	52.3 ± 2.26	55.6 ± 1.34	68.6 ± 1.17	85.7 ± 1.44	81.1 ± 1.03
ST-GCN [77]	55.3 ± 0.48	54.4 ± 1.03	73.2 ± 0.77	85.5 ± 0.44	82.6 ± 1.40
A3TGCN [74]	57.5 ± 0.75	61.3 ± 0.57	77.4 ± 0.96	85.6 ± 1.12	84.0 ± 0.16
ELMOBiLSTM [18]	50.6 ± 0.83	53.1 ± 0.22	70.5 ± 1.55	84.2 ± 0.99	81.8 ± 1.42
E-FCN [19]	61.6 ± 0.34	69.9 ± 1.11	78.5 ± 1.31	90.7 ± 0.33	84.6 ± 0.73
Bi-LSTM [17]	27.5 ± 2.05	45.5 ± 2.69	77.5 ± 1.56	90.1 ± 0.89	84.2 ± 0.47
TLGAT [21]	72.8 ± 0.63	74.5 ± 1.09	83.4 ± 1.72	91.8 ± 0.17	85.1 ± 0.83
Our GNN Approach	78.3 ± 0.95	80.4 ± 0.52	88.7 ± 0.60	92.4 ± 0.34	88.7 ± 0.22

**Table 3 sensors-24-03944-t003:** Classification performance on samples with a fixed set of left-out sensors (Setting 2) or random missing sensors (Setting 3) on several CASAS datasets. *Set. 2 F1* stands for F1 score obtained in Setting 2 and *Set. 3 F1* stands for an F1 score obtained in Setting 3. We report the mean and standard deviation of F1 scores across the runs.

Missing Sensor Ratio	Approach\Datasets	Milan		Kyoto8		Kyoto7	
		Set. 2 F1	Set. 3 F1	Set. 2 F1	Set. 3 F1	Set. 2 F1	Set. 3 F1
0%	BiLSTM [17]	45.5 ± 1.42	45.5 ± 1.42	27.5 ± 2.05	27.5 ± 2.05	77.5 ± 1.56	77.5 ± 1.56
	TLGAT [21]	74.5 ± 1.09	74.5 ± 1.09	72.8 ± 0.63	72.8 ± 0.63	83.4 ± 1.72	83.4 ± 1.72
	Our GNN Approach	80.4 ± 1.23	80.4 ± 4.68	78.3 ± 0.95	78.3 ± 0.95	88.7 ± 0.60	88.7 ± 0.60
10%	BiLSTM [17]	31.7 ± 2.48	32.1 ± 5.15	21.3 ± 2.11	26.2 ± 2.19	66.4 ± 1.26	69.2 ± 3.56
	TLGAT [21]	69.7 ± 1.89	70.2 ± 4.34	67.9 ± 1.11	63.8 ± 2.53	81.3 ± 0.93	82.0 ± 2.63
	Our GNN Approach	75.5 ± 2.85	77.6 ± 4.74	75.3 ± 1.96	71.3 ± 3.96	85.8 ± 0.58	86.4 ± 2.04
20%	BiLSTM [17]	24.4 ± 3.62	30.5 ± 4.93	12.0 ± 2.46	20.1 ± 2.45	50.1 ± 1.34	61.7 ± 3.78
	TLGAT [21]	67.1 ± 1.73	67.0 ± 2.13	55.6 ± 1.33	60.4 ± 3.11	80.8 ± 1.42	85.8 ± 1.98
	Our GNN Approach	71.3 ± 2.36	72.4 ± 4.42	65.2 ± 2.33	67.4 ± 3.87	85.6 ± 0.99	86.1 ± 0.39
30%	BiLSTM [17]	17.1 ± 2.95	22.7 ± 4.98	10.3 ± 1.26	12.2 ± 3.74	44.3 ± 1.97	59.1 ± 2.70
	TLGAT [21]	62.2 ± 2.41	65.1 ± 3.38	50.0 ± 1.87	51.1 ± 2.99	80.4 ± 0.88	84.5 ± 0.91
	Our GNN Approach	67.8 ± 2.86	67.6 ± 4.87	58.5 ± 2.01	55.5 ± 4.56	84.2 ± 1.66	84.5 ± 0.67
40%	BiLSTM [17]	8.0 ± 3.42	11.7 ± 5.18	8.1 ± 2.33	12.2 ± 4.67	35.1 ± 1.32	57.8 ± 0.89
	TLGAT [21]	53.7 ± 3.02	53.5 ± 4.26	42.4 ± 1.67	40.5 ± 4.73	79.8 ± 2.07	80.9 ± 0.24
	Our GNN Approach	53.8 ± 2.17	62.6 ± 4.88	47.5 ± 1.95	40.5 ± 4.95	79.9 ± 2.17	80.9 ± 0.17

**Table 4 sensors-24-03944-t004:** Ablation studies on CASAS datasets: Kyoto8, Milan, Kyoto7, Aruba, and Cairo. The baseline is a 2-layer fully connected network comparable to the graph in terms of the network size (measured by the number of parameters). The graph approach involves the use of the neural message-passing framework (outlined in Section 2.4). Embedding refers to adding the embedding layer (outlined in Section 3.3). *Attn* refers to the inclusion of the attention mechanism (outlined in Section 3.4.2). We report the mean and standard deviation of F1 scores across five runs.

Methods\Datasets	Kyoto8	Milan	Kyoto7	Aruba	Cairo
	F1 Score	F1 Score	F1 Score	F1 Score	F1 Score
Baseline	20.2 ± 2.47	41.3 ± 1.71	54.9 ± 1.84	66.7 ± 3.07	72.6 ± 0.88
Graph	71.5 ± 0.62	70.9 ± 0.89	76.8 ± 0.83	83.7 ± 1.68	78.4 ± 0.84
Graph + Embedding	73.5 ± 0.24	72.2 ± 2.11	80.6 ± 0.09	87.3 ± 0.94	82.2 ± 1.11
Graph + Embedding + Attn	78.3 ± 0.95	80.4 ± 0.52	88.7 ± 0.60	92.4 ± 0.34	88.7 ± 0.22

## Data Availability

The data used in the evaluation and training of models are openly available at https://casas.wsu.edu/datasets/. Accessed on 15 January 2022.

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
