# Peer review of "Using Graphs to Perform Effective Sensor-Based Human Activity Recognition in Smart Homes"

_sensors, 2024, doi:10.3390/s24123944_

Round 1
Reviewer 1 Report
Comments and Suggestions for Authors
The authors write that the article is novel due to the “novel and more expressive graph-based activity recognition system for smart homes that better models inter-sensor relationships;”. However, this should be clarified both qualitatively that quantitatively.
Why “A discussion on how to address an irregular sampling of sensor data streams as it is prevalent in smart homes” should be considered a key contribution?
Since the authors are describing also the main sensors exploited for human activity recognition, I suggest to mention radar technology which is widely used for monitoring the human activities, e.g., E. Cardillo, C. Li and A. Caddemi, "Radar-Based Monitoring of the Worker Activities by Exploiting Range-Doppler and Micro-Doppler Signatures," 2021 IEEE International Workshop on Metrology for Industry 4.0 & IoT (MetroInd4.0&IoT), Rome, Italy, 2021, pp. 412-416, doi: 10.1109/MetroInd4.0IoT51437.2021.9488464., or for different applications, e.g., as occupancy sensor to control the heating, ventilation, and air conditioning system.
The authors should highlight how their procedure overcomes the already known scientific literature. The general description of the approach is quite verbose. The authors should summarize the main step and above all, the differences compared to the existing approach.
Author Response
The authors write that the article is novel due to the “novel and more expressive graph-based activity recognition system for smart homes that better models inter-sensor relationships;”. However, this should be clarified both qualitatively that quantitatively.
Response: We believe our work addresses this prompt qualitatively under section 5.3. Can we explain the classifications performed by our proposed HAR system and quantitatively 4.2.3. Setting 3: Random sensors dropped. Section 5.3 speaks to how the networks are qualitatively more expressive as they provide more explainability into neural network based HAR approaches. Section 4.2.3 specifically shows empirical evidence of better relationships learned between sensors. Consequently, the network is robust to random sensor values getting dropped randomly - a real and practical scenario.
Why “A discussion on how to address an irregular sampling of sensor data streams as it is prevalent in smart homes” should be considered a key contribution?
Response: We have rephrased the writing to address this.
Since the authors are describing also the main sensors exploited for human activity recognition, I suggest to mention radar technology which is widely used for monitoring the human activities, e.g., E. Cardillo, C. Li and A. Caddemi, "Radar-Based Monitoring of the Worker Activities by Exploiting Range-Doppler and Micro-Doppler Signatures," 2021 IEEE International Workshop on Metrology for Industry 4.0 & IoT (MetroInd4.0&IoT), Rome, Italy, 2021, pp. 412-416, doi: 10.1109/MetroInd4.0IoT51437.2021.9488464., or for different applications, e.g., as occupancy sensor to control the heating, ventilation, and air conditioning system.
Response: We thank the reviewer for sharing some interesting works. However, we feel that the first work is out of the scope of this work since we are primarily focused on a network of smart home sensors to recognize common human activities in indoor settings instead of focusing on works that are primarily based on a single radar or wifi sensor.
The authors should highlight how their procedure overcomes the already known scientific literature. The general description of the approach is quite verbose. The authors should summarize the main step and above all, the differences compared to the existing approach.
Response: We explicitly outline our differences at the end of section 2 (background).
Reviewer 2 Report
Comments and Suggestions for Authors
This paper investigates the problem of how to use variable and sparse sensor measurements to achieve HAR in smart homes. The authors design a graph-based network to investigate the potential sequential relationships of sensor measurements and build the inter-sensor characters. The detailed case studies and discussions justify the effectiveness of the proposed method.
However, there are several weaknesses in the paper, which make the paper cannot be fully appreciated. The shortcomings are identified as follows.
1. On page 9, the author mentions that the hyperparameter k related to the number of edges is set to 5, but the results of empirical study for this choice is not introduced in the paper. To justify the rationality, the authors should introduce the process and results of empirical experiments. Additionally, they should discuss whether the value of k needs to be adjusted when the environment or sensors change.
2. In the experiments, a window size of 20 is used. The reason for this choice is not explained. Besides, and its potential impact on the algorithm's performance when the window size changes is not discussed. If the performance can be influenced by the sliding window's width, the statement "Our proposed approach - which does not require fixed windows or oracle specified time windows." is inaccurate and needs to be revised for clarity.
3. Appendix A only has a section title but lacks detailed explanation of aggregated sensor triggers. This section seems essential to justify the dataset's reliability of the experimental design, so the authors need to carefully complete this part. Besides, it seems Figure A1 is part of Appendix A. However, without specific explanations, it is difficult for readers to understand the author's intentions.
4. Besides data missing, sensor faults like data drift and noise are common but not addressed in the paper. It is unclear whether the current system remains effective if sensor values show significant drift or substantial noise.
5. This paper failed to properly cite several past literatures (e.g., [1-3]) highly related to this work, and clearly discuss the differences between them and this paper.
[1] RF-Based Human Activity Recognition Using Signal Adapted Convolutional Neural Network. TMC 2023
[2] ShoesLoc: In-Shoe Force Sensor-Based Indoor Walking Path Tracking, IMWUT 2019.
[3] ActListener: Imperceptible Activity Surveillance by Pervasive Wireless Infrastructures. ICDCS 2022
6. In addition, there are some writing issues in the paper. The authors may want to check them out to improve the presentation. For example,
a. On page 10, the variables W, x, and z are sometimes bolded and sometimes not. The authors should standardize the notation.
b. Table A2 is incomplete and some data are occluded.
c. The bottom line of Table 1 is missing.
d. The authors should provide an abbreviation for the proposed method. When referring to their method in tables, the authors always use blank text, which can confuse the readers.
e. The authors often include comparative analysis of table content in the table captions, which should be included in the main text instead. This writing style makes it harder to read the paper and leads to excessively long captions. To enhance the quality of the paper, the authors should simplify the captions.
Comments on the Quality of English Languagerefer to detailed comments
Author Response
We thank the reviewer for the feedback. We address each point below:
1. We have added a subsection to address the concern regarding the choice of hyperparameter k (Section 5.1 and Figure 5).
2. We do so to compare fairly and consistently with rest of the related works (Explained in section 4.1.2 Evaluation Methodology).
3. We have added more details to appendix A to clarify the key idea of the charts.
4. This is a valid concern which is reasonably addressed by our approach. This is empirically shown by high performance on the CASAS datasets. This set of data closely resembles real-world smart home sensor data including issues such as noise and data drift. Instead of explicitly trying to model data drift and noise, we chose a dataset that implicitly contains these issues and show that our model performs better than many prior state of the art methods.
5.
[1] - We have not compared this work because it is challenging to compare this work with the set of works we have studied for the following reasons: 1. the sensors involved are different (RF sensors are different from binary and temperature sensors in the setup we are studying) 2. The lack of availability of code and data makes it hard to make a fair comparison empirically with.
[2] - This is not a related work because the task being achieved in this work is too specific and not similar to the list of activities we are looking to predict.
[3] - We do not consider this a related work since the task of this work is different - the granularity of activity recognition is not the same as our work and all other related works we have compared against.
6. Thank you for pointing out the issues. We have addressed all of them.
Reviewer 3 Report
Comments and Suggestions for Authors
The manuscript presents a valiant effort in conducting comprehensive tracking in enclosed surroundings by means of graph-based learning approaches. There is a solid overview of the area and the experimental setup is decent. The results are admitted to be somewhat initial, in the sense that many more investigations can be undertaken within the same setup.
The main findings are very encouraging and point to great practicality of the proposed framework. It would seem that similar observations about the benefits of graph-based learning were made in previous work, but clearly not in this particular setup.
Author Response
The results are admitted to be somewhat initial, in the sense that many more investigations can be undertaken within the same setup.
Response: We agree more experiments can be conducted. However, we strongly believe that a novel architecture and several key findings have already been included in this work.
The main findings are very encouraging and point to great practicality of the proposed framework. It would seem that similar observations about the benefits of graph-based learning were made in previous work, but clearly not in this particular setup.
Response: Thank you for the comments. We have tried to replicate many other works in a similar setup as shown in Table 2.
Round 2
Reviewer 1 Report
Comments and Suggestions for Authors
the authors address my concerns
Reviewer 2 Report
Comments and Suggestions for Authors
comments addressed. recommend acceptance.
Comments on the Quality of English Languagena